# Contrast-Enhanced Imaging in the Management of Intrahepatic Cholangiocarcinoma: State of Art and Future Perspectives

**DOI:** 10.3390/cancers15133393

**Published:** 2023-06-28

**Authors:** Lucia Cerrito, Maria Elena Ainora, Raffaele Borriello, Giulia Piccirilli, Matteo Garcovich, Laura Riccardi, Maurizio Pompili, Antonio Gasbarrini, Maria Assunta Zocco

**Affiliations:** CEMAD Digestive Disease Center, Fondazione Policlinico Universitario “A. Gemelli” IRCCS, Università Cattolica del Sacro Cuore, 00168 Rome, Italy; lucia.cerrito@policlinicogemelli.it (L.C.); raffaele.borriello01@icatt.it (R.B.); giulia.piccirilli01@icatt.it (G.P.); matteo.garcovich@policlinicogemelli.it (M.G.); laura.riccardi@policlinicogemelli.it (L.R.); maurizio.pompili@unicatt.it (M.P.); antonio.gasbarrini@unicatt.it (A.G.); mariaassunta.zocco@unicatt.it (M.A.Z.)

**Keywords:** intrahepatic cholangiocarcinoma, contrast-enhanced computed tomography, magnetic resonance imaging, contrast-enhanced ultrasound

## Abstract

**Simple Summary:**

Contrast imaging techniques play a pivotal role in the diagnosis and management of Intrahepatic cholangiocarcinoma (iCCA). There is an increasing interest in the specific imaging features which can predict tumor behavior or histologic subtypes.

**Abstract:**

Intrahepatic cholangiocarcinoma (iCCA) represents the second most common liver cancer after hepatocellular carcinoma, accounting for 15% of primary liver neoplasms. Its incidence and mortality rate have been rising during the last years, and total new cases are expected to increase up to 10-fold during the next two or three decades. Considering iCCA’s poor prognosis and rapid spread, early diagnosis is still a crucial issue and can be very challenging due to the heterogeneity of tumor presentation at imaging exams and the need to assess a correct differential diagnosis with other liver lesions. Abdominal contrast-enhanced computed tomography (CT) and magnetic resonance imaging (MRI) plays an irreplaceable role in the evaluation of liver masses. iCCA’s most typical imaging patterns are well-described, but atypical features are not uncommon at both CT and MRI; on the other hand, contrast-enhanced ultrasound (CEUS) has shown a great diagnostic value, with the interesting advantage of lower costs and no renal toxicity, but there is still no agreement regarding the most accurate contrastographic patterns for iCCA detection. Besides diagnostic accuracy, all these imaging techniques play a pivotal role in the choice of the therapeutic approach and eligibility for surgery, and there is an increasing interest in the specific imaging features which can predict tumor behavior or histologic subtypes. Further prognostic information may also be provided by the extraction of quantitative data through radiomic analysis, creating prognostic multi-parametric models, including clinical and serological parameters. In this review, we aim to summarize the role of contrast-enhanced imaging in the diagnosis and management of iCCA, from the actual issues in the differential diagnosis of liver masses to the newest prognostic implications.

## 1. Introduction

Cholangiocarcinoma (CCA) represents a heterogeneous group of aggressive tumors arising from the epithelial cells at any point of the biliary tract. According to the site of origin, it can be classified into intrahepatic CCA (iCCA, when arising proximally to the second-order bile ducts, in the context of liver parenchyma), perihilar CCA (pCCA, originating between second order bile ducts and the insertion of the cystic duct into the common bile duct), and extrahepatic or distal CCA (dCCA, below the insertion of common bile duct) [1].

Another classification proposed by the Liver Cancer Group of Japan categorizes CCA according to the morphologic growth pattern into mass-forming CCA, periductal infiltrating CCA, and intraductal growing type [2].

These differences have not only an anatomic meaning but influence the whole clinical management and therapeutic approach with important implications in prognosis and survival rates [3,4].

In particular, iCCA represents the most lethal tumor among these categories and shows the highest rising incidence in Western countries during the last decades [3]. At present, it is the second most common liver cancer after hepatocellular carcinoma (HCC), accounting for about 15% of primitive liver malignant neoplasms [5,6]. However, it is estimated that its incidence could further grow up to 10-fold worldwide during the next years [7].

The poor prognosis and the higher mortality rate of iCCA compared to pCCA and dCCA are mainly due to the unspecific clinical manifestations and the consequent high percentage of late diagnosis [3,8].

Surgical resection is the only potentially curative therapy for iCCA, resulting in a 5-year survival rate ranging from 15 to 40% [9,10]. However, in most cases, the diagnosis occurs when the disease is already unsuitable for surgical treatment due to vascular or lymphatic invasion, contiguous abdominal structure involvement, multifocal disease, or metastatic spread [4,11,12]. Thus, the overall 5-year survival for patients with iCCA is estimated to be around 8–18% [8,13].

Considering the poor prognosis and the increasing spread of this disease, contrast-enhanced imaging plays a pivotal role in the early diagnosis and assessment of the disease, and growing attention is focused on the new imaging biomarkers of malignancy [14,15].

The main international guidelines state that computed tomography (CT) and/or magnetic resonance imaging (MRI) are the principal imaging modalities for iCCA diagnosis and lead to the identification of radiological patterns of iCCA, which could provide prognostic and therapeutic information, such as surgical resectability or systemic therapy response [16,17,18].

Systematic surveillance for CCA through radiological imaging decreases the risk of overall mortality and improves survival after diagnosis [3].

The population identified as at risk for developing CCA comprises a wide group of patients: primary sclerosing cholangitis (PSC) represents the main risk factor for CCA in the Western world [17]; other conditions are biliary diseases (choledochal cysts, choledocholithiasis, cholelithiasis, Caroli disease), chronic liver diseases (chronic hepatitis B or C), cirrhosis, hemochromatosis, or non-alcoholic fatty liver disease (NAFLD) [17]. Infestation from liver flukes, as well as other extra-hepatic or environmental conditions (inflammatory bowel disease, chronic pancreatitis, alcohol consumption, and exposition to toxic-like thorotrast, 1,2-dichloropropane, or asbestos), represent other acknowledged risk factors for CCA [17]. Although specific surveillance programs for CCA have not been developed for all the mentioned conditions, there is general agreement regarding the suggestion of performing annual radiologic surveillance in patients with PSC [17,19]. Moreover, the diagnosis of CCA can be extremely challenging: MRI combined with magnetic resonance cholangiopancreatography (MRCP) grants the highest diagnostic accuracy and is suggested for CCA surveillance, even if there is concern regarding the long-term effects of repeated gadolinium injections and the high costs [17,19]. A surveillance program for CCA is also recommended for patients affected by liver flukes’ infestation through a semestral abdominal ultrasound examination since this technique demonstrated favorable results in early diagnosis and improved survival in these subjects [17]. In patients with liver cirrhosis, a semestral abdominal ultrasound examination, which is already recommended for HCC surveillance, is also suggested for the early detection of CCA [17].

However, iCCA can express different radiological features, sometimes differing completely from the expected patterns and even mimicking hepatocellular carcinoma (HCC) in the case of nodules developed on the cirrhotic liver [20,21].

In the last years, besides standard radiological modalities, a growing interest has been focused on contrast-enhanced ultrasound (CEUS) and on the use of radiomics to obtain prognostic information [22].

The aim of this review is to summarize the state of art of the use of contrast-enhanced imaging for the diagnosis and management of iCCA, from the actual issues in the differential diagnosis of liver masses to the newest prognostic implications.

### 1.1. Computed Tomography

A multiphasic acquisition consisting of the pre-contrast phase and post-contrast arterial, portal, and delayed phases is essential to obtain a complete CT evaluation. Although there are no pathognomonic CT characteristics for iCCA identification, some imaging features may strongly indicate this diagnosis [23].

Classic CT features of iCCA include hypodense hepatic lesion without a capsule with dilatation of distal biliary ducts and, sometimes, capsular retraction due to the fibrotic nature of the tumor (Figure 1 and Figure 2).

The contrast enhancement pattern of the lesion is important, especially for the differential diagnosis with HCC: iCCA receives its blood supply from portal vein branches and, for this reason, presents portal or delayed-phase enhancement pattern, whereas HCC blood supply comes from hepatic arteries with a consequent arterial-phase enhancement pattern [24]. The principal distinctive feature of iCCA, due to the abundant cellularity in the peripheral parts of the neoplastic mass, is represented by the initial rim enhancement or the peripheral enhancement in the arterial phase with subsequent centripetal enhancement in the delayed phase [25]. This phase, which starts about 3–5 min after contrast agent injection, is essential for iCCA diagnosis and is characterized by a marked central enhancement due to the presence of abundant fibrotic tissue [23].

Even if, in the past years, the multiphasic contrast-enhanced CT scan was considered substantially equivalent to MRI in the detection of iCCA and showed a good predictive value in the evaluation of resectability [7,26,27,28], a recent study evidenced that its sensitivity is inferior to MRI in the detection of mass-forming iCCA [29]. For this reason, the latest European Association for the Study of the Liver (EASL) guidelines suggest performing MRI for local staging [17].

The evaluation of the iCCA enhancement pattern at the CT scan seems to be a valuable non-invasive prognostic index essential in treatment planning. In a retrospective study performed on 147 patients with iCCAs, Park et al. observed that the hypervascular lesions had better recurrence-free survival and post-recurrence prognosis compared to hypovascular tumors (hazard ratio [HR]: 6.241; 95% confidence interval [CI]: 2.670–14.586, *p* < 0.001) and rim-enhancement (HR: 3.893; 95%CI: 1.700–8.915; *p* = 0.001) [30].

A CT scan can also be applied to the detection of both lymph nodal and distant metastases, thus granting a correct assessment in the resectability evaluation, with an accuracy (88%) superior to that of MRI [31] and a negative predictive value ranging between 85 and 100% [32]. Olthof et al. observed a better sensitivity of CT in the detection of vascular involvement and extrahepatic spread, but, unfortunately, the literature on the role of CT scan in the staging of iCCA is quite poor [33]. In a recent multicenter study on 334 patients, Kim et al. compared CT and MRI for iCCA staging: no differences were detected in the primary lesion assessment (T stage), whereas CT was better in the identification of capsular or vascular infiltration. Conversely, previous studies demonstrated similar performances by the two radiological methods in terms of vascular invasion detection [29,34]. The authors demonstrated a higher specificity of CT compared to MRI in lymph nodal staging (N stage) and similar sensitivities (about 60%), even if both results were quite limited [29,35].

Noji et al. demonstrated that in CCA, the CT evaluation of the extent of paraaortic lymph nodal metastases reflected the presence of distant metastases and proposed diagnostic criteria for paraaortic lymph nodes detection (CT-determined lymph nodal dimension, shape, and internal structure). However, the positive predictive value was extremely low, ranging from 13% to 36% [36]. A better performance was obtained by Bartsch et al., who underlined the role of preoperative imaging (both CT and MRI) in the detection of suspicious lymph nodes and the subsequent prediction of resectability, disease recurrence, and long-term outcome. Unfortunately, the low sensitivity (71.1%) of preoperative imaging alone required exploratory laparoscopy to better define tumor resectability [37]. An important step forward was achieved by Zhu et al., who found consistent differences in the comparison of preoperative CT features of iCAA with or without lymph nodal metastases (arterial and portal phase enhancement degree-mean, arterial and portal phase enhancement degree-max, equilibrium phase [EP] enhancement ratio, EP CT value max, EP CT value max/liver). The multivariate logistic regression analysis identified three factors (morphologic classification, EP enhancement ratio, EP CT value max) as independent risk factors for lymph nodal metastases in patients with iCAA. Based on these factors, the authors built a nomogram that could be useful in predicting the presence of metastatic lymph nodes and overall survival, with better performance than the simple radiologic N staging [38].

The metastatic spread of iCAA could also involve distant organs, such as bones (14%), peritoneum (18%), and lungs (24%): in this context, CT plays a key role in the identification of extrahepatic metastases in both basal assessment and follow-up restaging through a complete examination of thorax, abdomen, and skeleton [39].

An important contribution to the evaluation of macrovascular invasion is offered by postprocessing techniques (e.g., multiplanar reconstruction) that allow an accurate analysis of the relationship of the neoplastic mass with the surrounding organs and vascular structures [15].

In a recent paper, Wakiya et al. elaborated on a CT-based deep-learning algorithm with high power in predicting post-surgery early recurrence, thus allowing to maximize the therapeutic performance. In a group of 41 patents with iCCA who underwent curative resection, this algorithm was able to predict recurrence with 97.8% sensitivity, 94.0% specificity, 96.5% accuracy, and an area under the receiver operating characteristic curve (AUROC) of 0.994 [40].

Similar results were obtained by Jiao et al. in 179 patients with mass-forming iCCA candidates for surgical resection [41]. In particular, they established two CT-based nomograms able to predict survival before and after resection, with increased accuracy compared to the most common prognostic patterns. The imaging-based preoperative nomogram included carcinoembryonic antigen (CA) 19-9 preoperative serum levels and specific imaging characteristics that were independent factors for scarce prognosis: the presence of multiple neoplastic nodules, the pattern of arterial enhancement, capsular retraction, and CT-detected lymph node metastases. The postoperative nomogram included all the previous elements and three histopathological parameters (neoplastic grade of differentiation, lymph node involvement, and capsular invasion). The authors concluded that the application of these CT-based nomograms could be important for the future prospects of a patient-tailored therapeutic strategy.

### 1.2. Magnetic Resonance Imaging

MRI and MRCP represent the imaging techniques of choice for CCA diagnosis. It appears as a hypointense lesion on T1-weighted sequences and heterogeneously hyperintense on T2-weighted sequences with a central hypointensity related to the presence of fibrosis (Figure 3 and Figure 4) [42,43]. A target-like image is obtained on diffusion-weighted sequences with increased restricted diffusion in the peripheral zones of the lesion (with diffusion restriction on apparent diffusion coefficient-ADC) and less restricted diffusion in the central area (low-signal intensity in diffusion-weighted imaging-DWI) [44].

Variable enhancement patterns are observed after gadolinium administration due to the distribution of neoplastic cells (increased number of arterial vessels) and fibrotic areas (associated with reduced vascularization): early peripheral enhancement with a concentric filling or with a central scar; complete enhancement; early and marked enhancement with subsequent heterogeneous washout starting from the peripheral parts of the lesion [45]. A thin rim enhancement is sometimes detected in the gadolinium-enhanced late phase due to the reduced arterial supply, while washout is identifiable in a congested part of the liver surrounding the tumor, where sinusoids are dilated [45]. In case of the abundant presence of neoplastic cells and increased arterial perfusion, early enhancement and late phase washout in the periphery of the lesion can be detected [45].

After gadoxetic acid enhancement, iCCA may present a target-shaped mass (central hyperintensity and peripheral hypointensity) or a lobulated shape with a weak surrounding rim [46]. Koh et al. evidenced the possible role of gadoxetic acid enhancement in the MRI hepatobiliary phase as a prognostic factor for mass-forming iCCAs due to its connection with the amount of intralesional fibrosis and neoplastic aggressiveness [47].

Similar results were previously obtained by Kajiyama et al. [48]. They demonstrated that a large amount of fibrosis in mass-forming iCCA is associated with a significantly poor prognosis: survival rates at 1 and 3 years were 64.3% and 22.0% in scirrhous-type iCCAs compared to 72.8% and 55.8% in nonscirrhous-type iCCAs, respectively. The multivariate analysis identified elevated proliferative activity of neoplastic cells, lymphatic permeation, perineural invasion, and lymph nodal metastatisation as independent prognostic factors related to poor prognosis. Moreover, the detection of abundant stromal fibrosis was useful for the assessment of iCCA biological aggressiveness.

According to Kang et al., high enhancement in the hepatobiliary phase is more common in moderately differentiated than in scarcely differentiated iCCA and is associated with reduced frequency of lymph node neoplastic involvement with subsequently better prognosis [49]. They also observed the sign of a “gadoxetic acid cloud” during hepatobiliary sequences, consisting of a poorly defined hyperenhancement in the central part of mass-forming iCCA determined by the enhancement of contrast agent in the fibrotic areas. Finally, they demonstrated that the hepatobiliary phase is crucial for the identification of satellites and intrahepatic metastases, both linked to a worse prognosis.

Mamone et al. retrospectively analyzed the aspect of 29 mass-forming iCCA during the MRI dynamic phase after gadobenate dimeglumine (Gd-BOPTA) administration: 93% of lesions had peripheral rim-like enhancement in arterial and portal–venous phase, with progressive filling at delayed phases [50]. A target pattern was identifiable in 56% of subjects. During the hepatobiliary phase, 96% iCAAs presented “cloud enhancement” (diffuse, inhomogeneous central enhancement), and in 79% of patients, the authors observed the association of “cloud enhancement” with a peripheral hypointense rim (target pattern). Unfortunately, these findings are not specific and can be identified even in other neoplastic lesions, such as adenocarcinoma metastases [50]. For this purpose, Xing et al. tried to define the MRI features predicting mass-forming iCCA in 43 patients with histopathological diagnosis and different degrees of differentiation: the presence/absence of intrahepatic metastases; the grade of the integrity of the hypointense ring in the hepatobiliary phase; and the target sign in T2-weighed sequences were highlighted as independent factors in the identification of differentiation degree of mass-forming iCCAs [51].

Several attempts were also made to implement MRI ability in distinguishing iCCA from HCC. Asayama et al. reported that the absence of both fibrous capsule and fat and the enhancement of gadoxetic acid at three minutes were more related to mass-forming iCAA than to poorly differentiated HCC [52]. More recently, Wu et al. created a multivariate logistic regression model based on specific characteristics of iCCA on gadoxetic acid-enhanced MRI: rim-like hyperenhancement in the arterial phase; delayed central enhancement; and targetoid pattern in the hepatobiliary phase [53]. These features together were able to distinguish iCAA from HCC, with a better diagnostic performance than targetoid appearance alone detected on CT and MRI (87.8% sensitivity, 92.21% specificity, 94.6% accuracy, and 0.94 area under the curve [AUC]).

Contrast-enhanced MRI seems to have an increasingly important role in predicting treatment response, with important implications in the definition of the better therapeutic choice. Ma et al. proposed the combination of quantitative and qualitative data from pre-surgical MRI (morphology of the iCCA, intrahepatic dilatation of the biliary ducts, maximum diameter of the lesion, visible sign of hepatic artery penetration, enhancement pattern during arterial phase, arterial phase edge enhancement ratio) with CA 19-9 levels and histological tumor grade as possible independent risk factors for the prediction of microvascular invasion in a preoperative setting [54]. On the other hand, Min et al. proposed the MRI enhancement pattern as a prognostic index for surgical patients [55]. They retrospectively examined the pre-operative MRI of 134 iCCA (mainly mass-forming tumors) who underwent curative resection. The 5-year risk of death was 5.9% for patients with diffuse hyperenhancement in the arterial phase, 87.9% in case of diffuse hypoenhancement (HR [95% CI]: 41 [5–312], *p* = 0.01), and 59.2% in patients with peripheral rim enhancement (HR [95% CI]: 11 [2–85], *p* = 0.02). The 5-year recurrence rate was 25.6%, 85.1%, and 79.4% in the case of diffuse hyperenhancement, diffuse hypoenhancement, and peripheral rim enhancement, respectively. The authors demonstrated that diffuse hyperenhancement was associated with a reduced frequency of vascular invasion and necrosis and a higher frequency of underlying chronic liver disease.

Moreover, the association of diffuse hyperenhancement in the arterial phase with uniform enhancement in the delayed phase has demonstrated a better prognostic value than the arterial hyperenhancement pattern alone [56].

The retrospective multicentre study by Kim et al., also reported by the recent EASL guidelines, demonstrated a superior sensitivity (but inferior specificity) of MRI compared to CT for the diagnosis of iCCA in all stages and the ability to identify multiple tumors, vascular, and visceral peritoneum invasion [17,29]. However, MRI with hepatobiliary contrast agents (gadoxetic acid- or gadobenate dimeglumine) allows better detection of intrahepatic metastases in poorly or undifferentiated tumors, with a crucial role in the decision of patient eligibility for curative resection. In N staging, MRI and CT had similar sensitivity (60%), but MRI presented a reduced accuracy for the pre-surgery prediction of lymph nodal metastasis [29,57].

Contrast-enhanced MRI has an important role in the prediction of response to systemic treatment, thus foreshadowing the possibility of selecting the ideal candidates for a specific treatment. In the retrospective study by Sheng et al., homogeneity, tumor margins, and peritumoral enhancement in the arterial phase have been identified as independent predictive factors of response to therapy (odds ratio [OR]: 14.93, 5.004, 5.076, and *p* = 0.019, 0.014, 0.042, respectively) [58].

EASL guidelines state that particular MRI sequences (DWI) are not reliable in the diagnostic differentiation of iCCA from HCC [59,60], but DWI can be useful in the identification of small metastases by iCCA [29]. The combination of DWI with the hepatobiliary phase could also be applied in the differential diagnosis of iCCA (mass-forming type) from atypical liver abscesses due to high sensitivity and high accuracy [61]. Moreover, Jiag et al. suggested that the combination of FDG PET/CT and abdominal MRI might improve the diagnostic accuracy for ICC [62].

In the last decade, several studies highlighted DWI as an efficient biomarker in the identification and characterization of neoplastic lymph nodes in various tumors [63,64]. Despite the good results obtained in other organs, it was not possible to obtain the same results in patients with iCCA. Apparently, neoplastic lymph nodes have lower values of ADC compared to benign lymph nodes due to the presence in the latter of reduced cellularity, smaller nuclei and nucleus/cytoplasm ratio, and wide extracellular space [65]. Promsorn et al. investigated the role of DWI as an indicator of overall survival through the identification of neoplastic lymph nodes, peritoneal metastases, or multifocal intrahepatic lesions [66]. A preliminary study by Holzapfel et al. documented the promising propriety of DWI in the differential diagnosis of benign and malignant lymph nodes with an 83.3% sensitivity, 92.8% specificity, 66.7% positive predictive value, and 96.7% negative predictive value [67]. However, another study by Promsorn et al. did not detect differences in ADC between benign and malignant lymph nodes in subjects with iCCA [68].

Zhang et al. proposed an MRI texture signature based on CD8+ T cells density as a predictive method for the non-invasive detection of iCCA immunophenotyping (important for the prediction of response to immune checkpoint inhibitors) and for the assessment of overall survival [69]. The authors demonstrated that the “inflamed” immunophenotyping (characterized by a higher density of CD8+ T cells) had a more favorable prognosis than the “non-inflamed” type, with a survival rate at 5 years of 48.5% and 25.3%, respectively (*p* < 0.05).

Finally, MRCP has an important role in detecting the level and the length of biliary tree involvement by iCCA, similar to that of endoscopic retrograde cholangiopancreatography and percutaneous transhepatic cholangiopancreatography. Particularly, MRCP represents a pivotal method for the analysis of the anatomy of the biliary tree and its main anatomic variants, thus giving a precise pre-operative assessment that allows the reduction of procedure-related complications [70].

Particularly, MRCP is useful in subjects with complete biliary obstruction and offers three-dimensional reconstructions of the complete biliary system. Unfortunately, it requires a longer scan time and is extremely sensitive to motion artifacts [15].

### 1.3. Positron Emission Tomography with Fluorodeoxyglucose

Positron emission tomography with 18-fluoro-2-deoxy-d-glucose (18FDG-PET) is a radiologic technique that identifies neoplastic tissues based on their high glucose utilization. This technique is not adequate to provide anatomical data on the localization of a lesion: to overcome this limitation, PET/Computed Tomography (PET-CT) was developed through the combination of a full-ring PET scanner with a multidetector-row helical CT. PET-CT shows higher sensitivity than CT and MRI in the identification of metastatic lymph nodes. This method is applied in case of potentially resectable disease at CT or MRI because it can help to establish the feasibility of surgery via the refinement of lymph nodal and distant-metastases staging [26]. Moreover, it can contribute to post-treatment follow-up with a sensitivity of 85–95% in the detection of mass-forming iCCA. In the case of recurrent or metastatic CCA, PET (with or without CT) has 94% sensitivity and 100% specificity compared to 82% and 43%, respectively, of CT alone [71].

According to both EASL and AASLD guidelines, PET-CT is not adequate for primary tumor staging (T stage) due to its low accuracy but has a relevant role in the detection of iCAA metastatic lymph nodes (Figure 4) and distant metastases that are not always easily identifiable by CT or MRI (PET-CT has higher sensibility, but is less satisfactory in case of lymph nodes with a diameter inferior to 1 cm) and should be performed routinely in all patients with apparently resectable lesions [17,19]. Huang et al. documented that 18-FDG PET/CT was more effective in lymph nodal metastases identification than MRI, even if negative results could not constitute a certainty for the absence of lymph nodal metastases [57]. Regarding distant metastases, PET-CT is a reliable diagnostic method [72], but it is not useful to exclude their presence due to possible false negatives (low sensitivity and high specificity) [57]. Lamarca et al. underline that 18FDG-PET can help to detect neoplastic areas in the body, but sometimes they can also highlight false-positive findings [73]. Finally, Kim et al. reported the higher specificity and accuracy of PET-CT compared to contrast-enhanced CT in the assessment of regional lymph node metastases in patients with iCCA and extrahepatic CCA [29].

The 18FDG-PET detection capacity has a significant clinical impact in the planning of iCCA treatment [74] since the presence of distance metastasis or locoregional lymph nodes represents a contraindication to surgery [75]. Moreover, some studies reported that lymph node status was strongly associated with iCCA prognosis [76]. Park et al. detected a positive correlation between the detection of lymph node metastasis on 18-FDG PET/CT and a 1-year recurrence after curative resection [77]. The role of lymph node metastasis and the value of lymph node dissection are still unclear in iCCA. About 17% of patients with iCCA have unsuspected lymph node metastases at diagnosis. For this reason, patients with apparently resectable iCCA should undergo lymph node sampling by endoscopic ultrasound to identify lymph node metastases before surgery. In particular, a biopsy should be performed after PET if it is negative or inconclusive [17].

### 1.4. Contrast-Enhanced Ultrasound (CEUS)

Since its introduction in clinical practice, contrast-enhanced ultrasound (CEUS) played a significant role in the characterization of liver lesions due to its ability to detect tumor perfusion in real-time with higher spatial and temporal resolutions than other contrast-enhanced imaging modalities [78]. It also has the advantages of being cost-effective [79], easily repeatable even in a short time, and with no risk of contrast-induced nephropathy, differently from contrast-enhanced CT and MRI [80].

Despite being an operator-dependent technique, CEUS is suggested in the characterization of newly discovered focal liver lesions in non-cirrhotic livers if a CT scan or MRI were inconclusive or contraindicated [80]. It can be an effective tool to discriminate between a benign and a possible malignant lesion, thus facilitating the decision as to whether an ultrasound-detected focal lesion in a non-cirrhotic liver should need further radiologic investigation or direct surgery [80]. However, in the context of the diagnosis of ICC, CEUS is insufficient as the sole diagnostic modality and its benefit is controversial, especially in the presence of chronic liver disease [17,19]. Indeed, in cirrhotic livers, CEUS could be useful to assess the probability of malignancy, even if Galassi et al. pointed out the significant risk of misdiagnosing iCCA as HCC with CEUS compared to CT (52% vs. 4.2%, *p* = 0.009) or MRI (52% vs. 9.1%, *p* = 0.02) [81]. For this reason, in cirrhotic subjects, CT and MRI are still required for a more accurate diagnosis and staging [80], while CEUS could help in the selection of focal liver lesions more suitable for a biopsy or for monitoring changes in the enhancement pattern [80].

Despite a discrete number of studies regarding iCCA’s behavior at CEUS examination, the existence of a diagnostic contrastographic pattern able to identify this tumor is still debated, and a wide variety of features have been identified [80,82,83]. In the arterial phase, iCCA can show a rim-like hyperenhancement, a non-rim-like hyperenhancement (homogeneous or heterogeneous), and also an hypoenhancement [80]. This variability in CEUS arterial enhancement patterns has been related to the presence of underlying liver disease and to iCCA morphological subtypes. Particularly, the arterial non-rim-like hyperenhancement can be observed more commonly in iCCA with underlying liver cirrhosis or chronic hepatitis, while the arterial rim-like hyperenhancement and the arterial hypoenhancement is more frequent in patients without the chronic liver disease [82,83,84]. The non-rim-like hyperenhancement also seems to be influenced by tumor size, being more frequent in small tumors [82,84]. Finally, all the described enhancement behaviors can be observed in mass-forming types of iCCA, whereas periductal infiltrating iCCA and intraductal iCCA show more frequently a heterogeneous non-rim hyperenhancement and homogeneous non-rim hyperenhancement, respectively [80]. On the other hand, in the portal–venous and late phases, all iCCA subtypes present an early (<60 s) and marked washout [80,82,84,85]. This feature is actually considered the main diagnostic difference between iCCA and HCC in patients with cirrhotic backgrounds since HCC is usually characterized by mild and late washout [80,85,86,87,88] (Figure 5).

In order to identify factors potentially associated with specific iCCA enhancement patterns, Yuan et al. performed CEUS in 96 patients with histologically diagnosed iCCA [82]. Most of the patients presented a non-rim-like enhancement (60/96) in the arterial phase, with a higher incidence in patients with underlying liver disease (*p* = 0.001). Homogeneous arterial iperenhancement has also been associated with a higher density of microvessels and arteries and lower fibrosis and necrosis rate compared to lesions with inhomogeneous arterial iperenhancement. The authors concluded that lesion size, presence of chronic viral hepatitis or liver cirrhosis, arterial/microvascular density, and presence of fibrosis or necrosis were the principal elements affecting arterial phase enhancement patterns in CEUS.

Similarly, Xu et al. retrospectively examined 32 iCCA to find a correlation between CEUS enhancement patterns and the entity of tumoral cell proliferation at histological examination [89]. They observed that areas characterized by arterial hyperenhancement in CEUS were those containing an augmented density of neoplastic cells. In particular, intraductal-developing iCCA, a subtype with plenty of neoplastic cells, presented homogeneous or less frequently heterogeneous hyperenhancement.

Interestingly, a small retrospective study by Chen et al. observed a peculiar portal–venous-phase washout pattern consisting of a different grade of washout among the center and the periphery of the tumor, resulting in a rim-like venous hyperenhancement [90]. This feature was never described before and could be related to an infiltrating growth pattern that is considered highly diagnostic for iCCA, even if other malignant liver lesions (e.g., metastases) must be excluded.

Despite all the specific enhancement features described in these studies, the distinction between iCCA and HCC in the cirrhotic liver remains particularly difficult due to the characteristics of the specific lesions mimicking one another.

In order to systematize the interpretation of CEUS patterns with those obtained with CT and MRI, the CEUS Liver Imaging Reporting and Data System (LI-RADS) system was released by the American College of Radiology (ACR) in 2016 [91]. Among LI-RADS categories, LR-M includes lesions with probable or certainly malignant nature but not specifically classifiable as HCC (iCCA, but also HCC-CCA mixed tumors and other non-HCC liver tumors). The typical CEUS findings of LR-M are peripheral rim-like enhancement during the arterial phase and early-marked (<60 s) washout during the portal and late phases. However, the detection rate of peripheral rim-like enhancement of iCCA varies from 31.5 to 73.3% among different institutions and, in addition, some HCCs also showed early washout on CEUS, which would be classified as LR-M, thus complicating the diagnostic landscape [92,93].

Many attempts were made to achieve a non-invasive CEUS-based diagnostic path for iCCA. In a retrospective study performed on 228 patients with LR-M liver lesions (99 iCCA, 129 HCC), Huang et al. identified peripheral rim-like arterial phase hyperenhancement associated with elevated Ca 19-9 as indicative of iCCA [94]. In particular, rim-like arterial hyperenhancement alone was documented in 50.5% of iCCA and 16.3% of HCC, with an AUC of 0.70, a sensitivity of 70.4%, and a specificity of 68.8% for iCCA diagnosis. When the authors combined the arterial enhancement pattern with Ca 19-9 levels diagnostic power of CEUS increased, reaching an AUC of 0.82 and a sensitivity of 100% without a significant increase in specificity (63.9%).

In a recent retrospective study performed on 511 liver lesions detected in 269 cirrhotic patients, the CEUS LR-M category had a 91.3% sensitivity and 96.7% specificity in predicting iCCA diagnosis, with a negative predictive value of 99.6% (*p* < 0.001) [95].

Li et al. tried to assess the usefulness of the LR-M category in the differentiation of iCCA from HCC in patients with and without risk factors for liver malignancies. The adjustment of early washout to <45 s increased the specificity of this feature to 95.61% (*p* = 0.004), with a sensitivity of 96.33% (*p* = 0.945). The authors concluded that the application of LR-M criteria, after proper adjustments, led to a reduction of HCC misdiagnosed as iCCA (from 12.3% to 4.4%) [96]. Similar results were obtained by Wildner et al. in a multicentre study [97] (Table 1).

Despite these optimistic results, LR-M class alone is not yet considered a valid diagnostic tool for iCCA because it could also include metastases, HCC, and rare liver malignancies [100].

A retrospective study by Chen et al. confirmed the importance of CEUS in the distinction between iCCA and HCC in situations at high risk for HCC [98]. They analyzed 88 patients with iCCA and 88 with HCC and developed a CEUS-based predictive model that proved to be superior to CEUS LI-RADS in iCCA identification. The score includes different CEUS features and, in particular, early washout, rim enhancement, the obscure boundary for intratumoral non-enhanced areas (representing copious fibrous/necrotic tissue in the central part of the lesion), marked washout, and intratumoral vein. This CEUS-based predictive model demonstrated higher performance than CEUS LI-RADS in the diagnosis of iCCA (AUC = 0.953 vs. 0.742; *p* < 0.001). However, the small number of enrolled patients, the absence of lesions different from iCCA or HCC, and the lack of comparison with standard radiological imaging decreased the statistical power of the study.

Recently, an attempt has also been made to distinguish iCCA from poorly differentiated HCC. Guo et al. retrospectively examined the US and CEUS features of 56 iCCA and 60 poorly differentiated HCCs; at univariate analysis, iCCA was characterized by irregular shape (related to its infiltrative growth), cholangiolithiasis, cholangiectasis, intratumoral vein, not-clear enhanced boundary, irregular rim enhancement in the arterial-phase, and early and marked washout in the portal and late phases [99]. On the other side, HCC was associated with a regular shape (due to expansive growth or the presence of a fibrous capsule), feeding artery, peripheral circular arteries, clear enhanced boundary, and slow washout. The authors observed a better differential diagnostic performance of US and CEUS signature (based on the previously reported features) compared to standard CEUS-LI RADS (AUC: 0.976, 0.955 and 0.758, sensitivity: 96.67%, 91.67% and 51.67%, and specificity: 92.45%, 90.57%, and 100% for US signature, CEUS signature and CEUS-LI RADS, respectively). Again, the relatively small number of examined cases, the possible selection bias related to the absence of both highly differentiated HCC and very poorly differentiated HCCs, and the absence of neoplastic lesions different from HCC and iCCA limited the diagnostic power of the study. However, the authors suggested using the additional US features, such as intratumoral vein and peripheral circular arteries, to increase the diagnostic accuracy of CEUS LI-RADS in distinguishing poorly-differentiated HCC from iCCA.

A recent meta-analysis performed by Chen and colleagues underlined the role of CEUS as a potentially valuable non-invasive tool for the differentiation of iCCA and HCC [87]. The authors analyzed in detail eight studies, including 1116 HCC and 529 iCCA, and identified three CEUS features that could identify HCC and iCCA (hyperenhancement in the arterial phase, mild and late washout in the portal–venous, and late phases for HCC and arterial rim-enhancement, marked and early washout in the portal–venous and late phases for iCCA).

The pooled diagnostic performance of CEUS in differentiating HCC from iCCA was good, with a specificity of 0.87 (95%CI: 0.79–0.92), a sensitivity of 0.92 (95%CI: 0.84–0.96), a positive likelihood ratio of 7.1 (95%CI: 4.1–12.0), a negative likelihood ratio of 0.09 (95%CI: 0.05–0.19), and AUROC of 0.95 (95%CI: 0.93–0.97) and an OR of 76 (95%CI: 26–220).

Another interesting field is represented by the prognostic value of the contrastographic patterns. In a retrospective study performed on 197 mass-forming iCCA, a non-rim-like arterial enhancement pattern has been associated with better overall survival than a rim-like one and was useful for predicting prognosis before surgery [83]. Similar results were obtained by Wang et al. in 29 patients with iCCA treated with microwave ablation; they identified rim-like arterial enhancement as an independent predictor of poorer prognosis and post-operative development of extrahepatic metastases [101].

### 1.5. Radiomics

Radiomics is an emerging field and a continuously evolving branch for tumor diagnosis, therapeutic decisions, and prognosis prediction. It deals with the extraction of quantitative data from high-quality medical images (defined as “radiomic features”), from which the radiologist identifies a region of interest (ROI) containing the entire neoplastic lesion or one of its subregions and finally presents it in two or three dimensions. Radiomic features about tissue and lesion characteristics (such as shape, grayscale, and texture) can be used alone or in combination with demographic, histologic, and genomic data in order to develop predictive or prognostic models. Radiomics is a tool that is planned to be used by clinicians as a guide in precise medical decisions [102,103].

This futurist methodology was also applied to iCCA, and considering the limitations of currently available imaging, it is now facing important challenges, such as preoperative prediction of lymph node metastases, microvascular invasion, and risk of early recurrence [103]. Radiomics could facilitate clinical decision-making and define the subgroup of patients who will benefit most from surgery or systemic therapies.

Regarding lymph node assessment, Zhang et al. built a radiomic-based nomogram (applicable in both CT and MRI) that provided favorable differential and prognostic results for iCCA after independent external validation [103]. Another study by Ji et al. developed a radiomics predictive model based on arterial phase CT images for the differentiation of patients at high-risk and low-risk for metastatic lymph nodes. The authors concluded that it could be a powerful diagnostic tool granting information about overall survival, but it is restricted by important limitations (single-centered study, absence of external validation cohort, features based only on CT arterial phase) [104].

Another specific risk factor in terms of prognosis in patients undergoing radical resection of iCCA is microvascular invasion (MVI). The preoperative prediction of MVI in iCCA is still difficult and principally detected by microscopic histopathology on resected surgical specimens. In the future, refined radiomics predictive models could grant better performances in terms of MVI detection through the combination of radiomics features, radiological imaging (US, CT, MRI), clinical information, and biomarkers [105,106,107]. Recent studies about radiomics investigated not only CT and MRI but also US and PET: a non-invasive preoperative prediction of MVI by iCAA could be provided by 18F-FDG PET/CT, with better results than CT alone or PET-combined CT, as demonstrated by Fiz et al. and Jiang et al. [108,109].

In their study on multi-organ tumors, Xu et al. found better predictive performances of radiomics signatures obtained by 3-dimensional features than 2-dimensional ones in iCCA at both univariate and multivariate analysis [110].

The predictive models built so far through radiomics are still limited due to low accuracy and poor reproducibility and are consequently not widely used in clinical practice. Due to its possible multifaceted applications, radiomics represents a great opportunity in the research field of tumors and, particularly, iCCA. For this reason, it will be necessary to standardize radiomic protocols and radiological scans in order to uniformize the imaging acquired by different medical centers and achieve accurate diagnostic and prognostic models [111].

## 2. Present Shadows and Future Perspectives

No specific radiological pattern for iCCA has been identified yet with the principal diagnostic techniques (Table 2), and the sensibility, specificity, and positive and negative predictive values of each imaging method, unfortunately, are still far from diagnostic accuracy in iCCA diagnosis and staging (Table 3). Thus, this tumor is still a widely unsolved challenge.

The recent guidelines by AASLD and EASL still recommend histological or cytological examination to confirm the diagnosis [17,19]. In the case of very small liver lesions, a biopsy could not be feasible, or the procedure itself may be burdened with heavy risks or result in a false negative, as happens in about 30% of cases [112]. For these reasons, despite the crucial importance of histological diagnosis of iCCA, the need for biopsy is controversial, and patients with smaller lesions may undergo treatment without histological confirmation [113]. However, surgery is reserved for a small group of patients and is charged with high rates of recurrence.

At present, the poor sensitivity and specificity of CT, MRI, and 18FDG-PET in lymph node detection represent a burden for a correct assessment of iCCA and the subsequent adequate therapeutic strategy.

The improvement of surveillance programs for the detection of early-stage liver tumors will probably lead in the next years to expanding the radiological diagnostic potential, resulting in more precocious surgical interventions (with higher rates of radical neoplastic resections). Moreover, the implementation of radiomics in clinical practice could gradually acquire a more prominent role alongside other futuristic diagnostic tools (e.g., liquid biopsy) and traditional imaging in both diagnosis and post-treatment surveillance.

**Table 2 cancers-15-03393-t002:** The main radiological features characterizing intrahepatic cholangiocarcinoma on computed tomography, magnetic resonance, and contrast-enhanced ultrasound.

Computed Tomography	Magnetic Resonance	Contrast-Enhanced Ultrasound
rim-enhancement in arterial phase centripetal enhancement in delayed phase portal or delayed-phase enhancement pattern hypodense hepatic lesions without a capsule, with dilatation of distal biliary ducts, and, sometimes, capsular retraction Delayed contrast enhancement	hypo/isointense on T1w sequences heterogeneously hyperintense on T2w sequences rim-like hyperenhancement in arterial phase rim-enhancement (sometimes) on gadolinium-enhanced late phase delayed central enhancement “EOB-cloud” in hepatobiliary phase targetoid pattern in hepatobiliary phase	arterial non-rim-like enhancement (cirrhosis or chronic hepatitis) arterial rim-like enhancement (no chronic liver disease) arterial hypoenhancement early (<60 s) washout marked washout rim-like venous hyperenhancement peripheral rim-hyperenhancement in portal phase and gradually decreasing enhancement in late phase

**Table 3 cancers-15-03393-t003:** An overview of the different studies analyzing the diagnostic accuracy of the main radiological imaging techniques for diagnosis and staging of intrahepatic cholangiocarcinoma.

	Number of Patients	Diagnostic Imaging	Primary iCCA	Metastatic Lymph Nodes	Distant Metastases
Kim et al. [74]	123	18FDG-PET/CT	Se = 84.0% Sp = 79.3% PPV = 92.9% NPV = 60.5% Accuracy = 82.9%	Accuracy = 75.9%	Accuracy = 88.3%
		CT	-	Accuracy = 60.9%	Accuracy = 78.7%
Kim YY et al. [29]	334	MRI	Se = 91.0% (T1b), 89.1% (T2), 77.8% (T3 or T4) Sp = 53.9% (T1b), 39.8% (T2), 45.5% (T3 or T4)	Se = 65.4% Sp = 54.6%	Vascular invasion: Se = 73.8% Sp = 47.0% Visceral peritoneal invasion: Se = 77.2% Sp = 45.1%
		CT	Se = 80.5% (T1b), 73.8% (T2), 58.0% (T3 or T4) Sp = 69.2% (T1b), 54.0% (T2), 63.2% (T3 or T4)	Se = 64.0% Sp = 36.4%	Vascular invasion: Se = 56.8% Sp = 60.3% Visceral peritoneal invasion: Se = 57.0% Sp = 62.38%
Park et al. [77]	18	CT	ND	Se = 20.0% Sp = 86.4%	ND
		18FDG-PET/CT	ND	Se = 80.0% Sp = 92.3%	ND
Lee et al. [114]	99	18FDG-PET/CT	Se = 90.2% Sp = 70.6% PPV = 93.7% NPV = 60.0% Accuracy = 86.9%	PPV = 94.1%	Se = 94.7%
		CT	Se = 84.2% Sp = 70.6% PPV = 93.2% NPV = 48.0% Accuracy = 81.8%	PPV = 77.5%	Se = 63.2%
Lin et al. [115]	291	PET-CT 18FDG-PET/CT	NA	Se = 83.0% Sp = 88.3% PPV = 81.6% NPV = 89.3% Accuracy = 86.3%	Se = 87.8% Sp = 95.4% PPV = 86.7% NPV = 95.8% Accuracy = 93.5%
Nishioka et al. [116]	202	CT	Se = 49% Sp = 100% PPV = 100% NPV = 75% Accuracy = 80%	Se = 40% Sp = 80% PPV = 63% NPV = 68% Accuracy = 67%	Macrovascular invasion: Se = 60% Sp = 89% PPV = 25% NPV = 97% Accuracy = 88% Bile duct invasion: Se = 17% Sp = 99% PPV = 25% NPV = 99% Accuracy = 93%
		MRI	Se = 51% Sp = 97% PPV = 92% NPV = 76% Accuracy = 79%	Se = 56% Sp = 83% PPV = 74% NPV = 76% Accuracy = 76%	Macrovascular invasion: Se = 60% Sp = 94% PPV = 38% NPV = 98% Accuracy = 92% Bile duct invasion: Se = 50% Sp = 99% PPV = 50% NPV = 96% Accuracy = 93%
		18FDG-PET/CT	Se = 29% Sp = 100% PPV = 100% NPV = 71% Accuracy = 72%	Se = 84% Sp = 86% PPV = 91% NPV = 84% Accuracy = 86%	Macrovascular invasion: Se = 40% Sp = 98% PPV = 50% NPV = 97% Accuracy = 94% Bile duct invasion: Se = 17% Sp = 100% PPV = 94% NPV = 94% Accuracy = 93%
Petrowsky et al. [117]	61	CT	Se = 78% Sp = 80% PPV = 92% NPV = 57% Accuracy = 79%	Se = 24% Sp = 86% PPV = 50% NPV = 65% Accuracy = 62%	Se = 25% Sp = 100% PPV = 100% NPV = 84% Accuracy = 85%
		PET-CT	Se = 93% Sp = 80% PPV = 93% NPV = 80% Accuracy = 89%	Se = 12% Sp = 96% PPV = 67% NPV = 64% Accuracy = 64%	Se = 100% Sp = 100% PPV = 100% NPV = 100% Accuracy = 100%
Vidili et al. [95]	269	CEUS (LiRADS-M)	Se = 91.3% SP = 96.7% PPV = 56.8% NPV = 96.5% Accuracy = 96.5%	NA	NA
Holzapfel et al. [67]	24	MRI (DWI + respiratory-triggered single-shot echo-planar imaging)	NA	Se = 83.3% Sp = 92.8% PPV = 66.7% NPV = 96.7%	NA
Lamarca et al. [73]	198	18FDG-PET	Se = 37.5% Sp = 97.0%	Se = 37.5% Sp = 97.0%	NA

Abbreviations: 18FDG-PET/CT = 18-fluoro-2-deoxy-d-glucose Positron Emission Tomography/Computed Tomography; DWI = diffusion-weighted imaging; iCCA = intrahepatic cholangiocarcinoma; MRI = magnetic resonance imaging; NA = not available; NPV = negative predictive value; PPV = positive predictive value; Se = sensitivity; Sp = specificity; CT = computed tomography.

## 3. Conclusions

The role of the US in the diagnostic workup of iCCA has been quite marginal so far due to its scarce diagnostic power compared to both CT scan and MRI. In this apparently well-defined landscape where CT and MRI still remain irreplaceable tools for iCCA diagnosis and staging, CEUS is gaining growing attention due to its possibility to produce a dynamic evaluation of the vascularization of the lesion, which provides a sort of identity mark of the tumor. In this context, it offers the future possibility to grant a rapid, reliable, cost-effective, and non-invasive diagnostic assessment for iCCA, up to its potential application in the accurate planning of the better therapeutic approach and in the prediction of treatment response and prognosis.

## Figures and Tables

**Figure 1 cancers-15-03393-f001:**
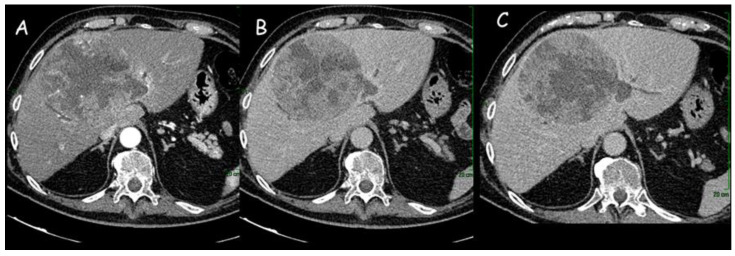
(**A**,**B**) Computed tomography imaging of intrahepatic cholangiocarcinoma (12 cm × 11 cm) of the right liver lobe: dishomogeneous enhancement in the arterial phase with hypodense areas of intralesional necrosis. (**C**) Dilatation of the peripheral intrahepatic biliary ducts.

**Figure 2 cancers-15-03393-f002:**
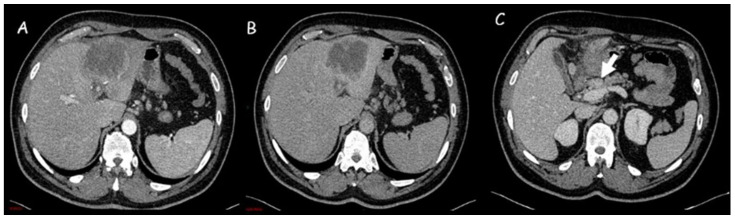
(**A**,**B**) Computed tomography imaging of an intrahepatic cholangiocarcinoma of approximately 8 cm in the left hepatic lobe, with capsular retraction, central hypodensity, and inhomogeneous enhancement due to necrotic-colliquative phenomena. Peripheral rim enhancement in the arterial phase (**A**) with progressive contrast filling in the subsequent phases (**B**). (**C**) Locoregional lymph nodes (white arrow).

**Figure 3 cancers-15-03393-f003:**
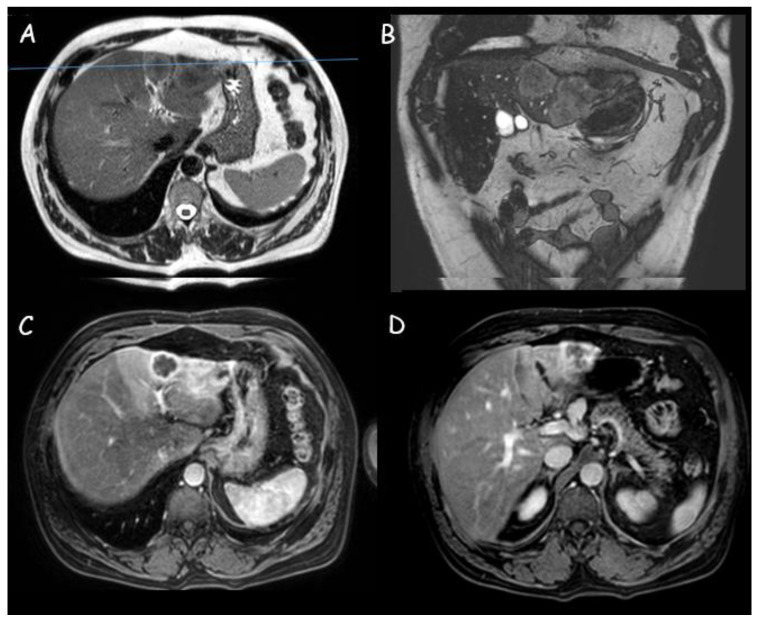
Magnetic resonance imaging of a 3 cm intrahepatic cholangiocarcinoma in segment III: (**A**,**B**) Moderately hyperintense lesion in T2 weighted sequences with weak restriction in diffusion-weighted imaging; (**C**,**D**) Peripheral contrast enhancement with perfusional alterations.

**Figure 4 cancers-15-03393-f004:**
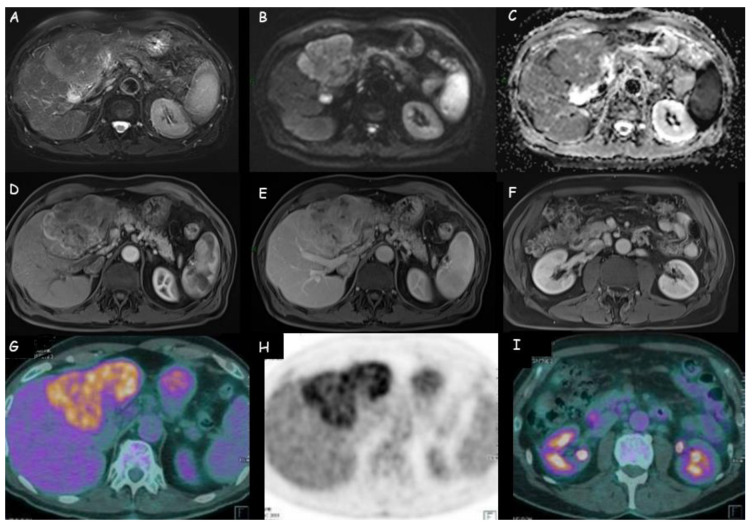
(**A**–**E**) Magnetic resonance imaging of an intrahepatic cholangiocarcinoma of 11 cm × 7 cm in the left liver lobe: weak hyperdensity in T2 weighted sequences; restriction at diffusion-weighted imaging (DWI); peripheral wash-in with late central retention. (**F**) Lymphadenopathies in celiac, hepatic, and left paraaortic area. (**G**,**H**) Positron emission tomography with 18-fluoro-2-deoxy-d-glucose (18FDG-PET) of the same lesion: increased uptake of the metabolic agent in the hepatic hypodense lesion, with markedly inhomogeneous distribution of the radiopharmaceutical. (**I**) Weak uptake of the metabolic tracer in the locoregional lymph nodes.

**Figure 5 cancers-15-03393-f005:**
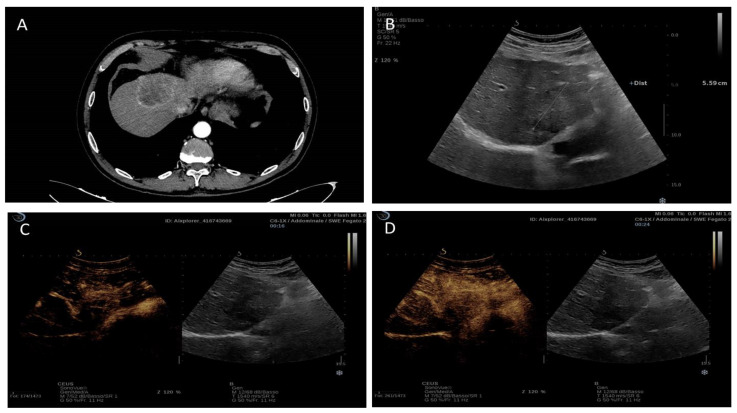
Contrast-enhanced imaging features a lesion of 5.5 cm intrahepatic cholangiocarcinoma of segment IV. (**A**) CT imaging: peripheral hyperenhancement in the arterial phase; (**B**) Hypoechoic aspect in B-mode ultrasound. (**C**,**D**) Contrast-enhanced ultrasound: peripheral hyperenhancement in the arterial phase and subsequently marked early washout.

**Table 1 cancers-15-03393-t001:** Contrast-enhanced ultrasound features of intrahepatic cholangiocarcinoma.

Study	Patients (n)	Arterial Phase (Hyperenhancement)	Portal/Late Phases	
Chen LD et al. [98]	88 HCC 88 iCCA	1.8% HCC 64.5% iCAA	Hyperenhancement: 98.2% HCC 88.7% iCCA Hypoenhancement: 11.3% HCC 1.8% iCCA	Early washout: 30.9% HCC 91.9% iCCA Marked washout: 1.3% HCC 61.3% iCCA
Chen T et al. [90]	21 iCCA	4.8%	Heterogeneous hyperenhancement: 61.9% Homogeneous hyperenhancement: 19.0% Isoenhancement: 4.8% Hypoenhancement: 4.8%	Rim-like venous hyperenhancement: 66.7% Hypoenhancement: 85.7% (portal–venous phase) 95.2% (late phase)
Guo et al. [99]	56 iCCA 60 HCC	Irregular 62.5% iCCA 3.3% HCC	Hyperenhancement: 94.6% iCAA 100% HCC	Washout in late phase: 94.6% iCCA Marked washout: 67.9% iCCA 6.7% HCC
Huang et al. [94]	99 iCCA 129 HCC	50.5% iCCA 16.3% HCC	15.2% iCCA 37.2% HCC	Early washout: 93.4% iCCA 96.1% HCC Marked washout: 23.2% iCCA 7.8% HCC
Li et al. [96]	With risk factors: 59 HCC 55 iCAA	0% HCC 42.6% iCCA Risk factors: cirrhosis, chronic hepatitis	100.0% HCC 50.0% iCCA	Early washout: 3.6% HCC 90.7% iCCA Marked washout: 3.4% HCC 79.6% iCCA
	Without risk factors: 55 HCC 55 iCCA	0% HCC 52.7% iCCA	100.0% HCC 45.5% iCCA	Early washout: 5.6% HCC 92.7% iCCA Marked washout: 9.1% HCC 89.1% iCCA
Wildner et al. [97]	42 iCCA 278 HCC Cirrhosis: 16.7% iCCA 76.9% HCC	85.7% iCCA 61% HCC	Center: 16.7% iCCA 60.3% HCC Periphery: 40.5% iCCA 75% HCC	Early washout (portal–venous phase): -Tumor center: 85.8% iCCA 49.8% HCC - Tumor periphery: 66.7% iCCA 32.6% HCC Washout in late phase: 92.9% iCCA 75% HCCs
Xu et al. [89]	32 iCCA	59.4%	Heterogeneous hyperenhancement: 18.8% Homogeneous hyperenhancement: 9.4% heterogeneous hypo-enhancement: 12.5%	Portal phase: -isoenhancement: 3.1% -hypoenhancement: 96.9% Late phase: hypoenhancement: 100%

Abbreviations: Intrahepatic cholangiocarcinoma, iCCA; Hepatocellular carcinoma, HCC.

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
