# Peer review of "Contrast-Enhanced Imaging in the Management of Intrahepatic Cholangiocarcinoma: State of Art and Future Perspectives"

_cancers, 2023, doi:10.3390/cancers15133393_

Round 1

Reviewer 1 Report

Dear Authors,

I would like to thank you for the opportunity to review this interesting paper focused on a very remarkable and challenging topic that is a lively argument also in daily clinical practice. 

Despite the recent technical advances in surveillance, the incidence and mortality rate of intrahepatic cholangiocarcinoma (iCC) have been rising during the last years and total new cases are expected to increase up to 10-fold during the next two or three decades. As no specific iCC radiology pattern exists, histopathological or cytological analysis is still mandatory to confirm the diagnosis. However, contrast-enhanced imaging techniques can help radiologists exclude other possible diagnoses, correctly stage the tumor and, especially by implementing new radiomic software, provide useful prognostic information.

This paper is beautifully written and pleasurable to read, although it suffers from some limitations that Authors can easily adjust to slightly improve their review making it more eligible for this important Journal. Furthermore, the Authors can improve some sections of the paper, adding information and including other important references about this topic that, in my opinion, should be cited and discussed. 

First of all, the title is clear and direct. Personally, from a stylistic point of view, I believe it could be improved and more focused on the scope of the review: “Contrast Enhanced Imaging in the management of Intrahepatic Cholangiocarcinoma: State Of Art And Future Perspectives”

Lines 65-67: Despite contrast-enhanced imaging playing a pivotal role in the early diagnosis and assessment of the iCC, no specific radiology pattern exists and, according to both American and European guidelines [doi: 10.1002/hep.32771] [doi: 10.1016/j.jhep.2023.03.010], histopathological or cytological analysis is still mandatory to confirm the diagnosis. Unfortunately, a liver biopsy is not always feasible, especially in the case of very small liver nodules; furthermore, this procedure is not free from risks and is limited by an elevated rate of false negatives (about 30%) [doi:10.3390/jcm11154399]. Therefore, the need for a biopsy is debated in patients with potentially resectable disease [doi: 10.1245/s10434-021-09671-y.]. In this uncommon situation, people will be treated for iCC based solely on imaging findings and other test results. In my opinion, with the now improved sensibility of surveillance programs and the increased detection of small liver cancers at early stages, the role of imaging will probably expand in the next few years and, also supported by the implementation of radiomics in clinical practice, will play an even greater role. Please briefly discuss and underline this topic and cite all the aforementioned references. These considerations could also be addressed in a new additional chapter before the conclusion, in which the current limitations of imaging techniques in iCC diagnosis and its future perspective and promises can be discussed.

Lines 74-75. Please briefly discuss which techniques are recommended to perform surveillance and which patients should be monitored [doi: 10.1002/hep.32771] [doi: 10.1016/j.jhep.2023.03.010].

Lines 103-104 “Even if both CT scan and MRI are equivalent in the identification of both primary lesions and satellites”. EASL guidelines state that “MRI should be considered instead of CT scanning for staging iCC within the liver” (strong recommendation) [doi: 10.1016/j.jhep.2023.03.010] since MRI was superior to CT in staging iCC for T1B, T2, and even T3/T4 tumours [doi: 10.1002/hep4.1774.]. Please revise this sentence.

Lines 232-234. MRCP is pivotal also to understanding the anatomy of the biliary tree and its main variations and thus providing precise pre-operative information for surgeons in order to reduce the rate of procedure-related complications [doi: 10.1111/joa.13808].

Lines 244-250. It is important to state that CEUS is an operator-dependent technique.

Lines 251-252. The affirmation that “CEUS is recommended as the first-line imaging technique in the characterization of newly discovered focal liver lesion in non-cirrhotic livers” is not true. According to EASL guidelines [doi: 10.1016/j.jhep.2023.03.010], despite the first suspicion of iCC is usually raised on ultrasound, the benefit of contrast-enhanced ultrasound (CEUS) in iCC is controversial and this, as correctly stated, is especially true in the presence of underlying chronic liver disease [doi: 10.1111/liv.12124]. Similarly, according to AASLD guidelines, “Contrast‐enhanced US, although insufficient as the sole diagnostic modality, may be considered when CT or MRI is inconclusive” [doi: 10.1002/hep.32771]. Moreover, as reported by the latest EFENSUB guidelines Guidelines for Contrast Enhanced Ultrasound (CEUS) in the Liver [doi: 10.1055/a-1177-0530], CEUS is recommended for the characterization of focal liver lesions in the non-cirrhotic liver in patients with inconclusive findings at CT or MR imaging (strong recommendation) or if both CT and MR imaging are contraindicated (strong recommendation). The primary aim of CEUS in patients with a non-cirrhotic liver is to differentiate benign from malignant and, therefore, it is useful to facilitate the clinical decision as to whether a sonographically detected liver lesion needs further investigation (CT or MRI) or surgery. Please revise this concept.

Please provide a Table reporting the imaging features of iCC with every discussed imaging technique.

Regarding Figure 1. Please provide an example of iCC in every imaging technique (CT, MRI, CEUS and PET), making sure to provide both arterial and venous phase images and, if possible, also those MRI sequences with ancillary findings that are highly suspicious for iCC (as hyperintensity in T2-w images and restricted diffusion).

Please also expand the role of PET scanning in iCC management, further underlying its usefulness in identifying lymph node metastasis.

From a stylistic point of view, I suggest discussing the current evidence regarding radomics in a single and separate chapter rather than discussing it in at the end of each chapter. Since its use in clinical practice has not yet been implemented, I believe these changes will facilitate reader comprehension               

Finally, I think references should be reformatted as suggested by the Author’s guidelines (Author 1, A.B.; Author 2, C.D. Title of the article. Abbreviated Journal Name YearVolume, page range).

Kind regards,

Author Response

Dear Authors,

I would like to thank you for the opportunity to review this interesting paper focused on a very remarkable and challenging topic that is a lively argument also in daily clinical practice. 

Despite the recent technical advances in surveillance, the incidence and mortality rate of intrahepatic cholangiocarcinoma (iCC) have been rising during the last years and total new cases are expected to increase up to 10-fold during the next two or three decades. As no specific iCC radiology pattern exists, histopathological or cytological analysis is still mandatory to confirm the diagnosis. However, contrast-enhanced imaging techniques can help radiologists exclude other possible diagnoses, correctly stage the tumor and, especially by implementing new radiomic software, provide useful prognostic information.

This paper is beautifully written and pleasurable to read, although it suffers from some limitations that Authors can easily adjust to slightly improve their review making it more eligible for this important Journal. Furthermore, the Authors can improve some sections of the paper, adding information and including other important references about this topic that, in my opinion, should be cited and discussed. 

First of all, the title is clear and direct. Personally, from a stylistic point of view, I believe it could be improved and more focused on the scope of the review: “Contrast Enhanced Imaging in the management of Intrahepatic Cholangiocarcinoma: State Of Art And Future Perspectives”.

Re: According to reviewer suggestion we have modified the title. 

Lines 65-67: Despite contrast-enhanced imaging playing a pivotal role in the early diagnosis and assessment of the iCC, no specific radiology pattern exists and, according to both American and European guidelines [doi: 10.1002/hep.32771] [doi: 10.1016/j.jhep.2023.03.010], histopathological or cytological analysis is still mandatory to confirm the diagnosis. Unfortunately, a liver biopsy is not always feasible, especially in the case of very small liver nodules; furthermore, this procedure is not free from risks and is limited by an elevated rate of false negatives (about 30%) [doi:10.3390/jcm11154399]. Therefore, the need for a biopsy is debated in patients with potentially resectable disease [doi: 10.1245/s10434-021-09671-y.]. In this uncommon situation, people will be treated for iCC based solely on imaging findings and other test results. In my opinion, with the now improved sensibility of surveillance programs and the increased detection of small liver cancers at early stages, the role of imaging will probably expand in the next few years and, also supported by the implementation of radiomics in clinical practice, will play an even greater role. Please briefly discuss and underline this topic and cite all the aforementioned references.

Re: We thanks the reviewer for raising this point. These considerations have been addressed in a new paragraph, in which the current limitations of imaging techniques in iCC diagnosis and future perspectives and promises have been discussed. (see lines 626-652)

Lines 74-75. Please briefly discuss which techniques are recommended to perform surveillance and which patients should be monitored [doi: 10.1002/hep.32771] [doi: 10.1016/j.jhep.2023.03.010].

Re: According to reviewer suggestion we added in the text more informations concerning iCC surveillance (see lines 74-96).

Lines 103-104 “Even if both CT scan and MRI are equivalent in the identification of both primary lesions and satellites”. EASL guidelines state that “MRI should be considered instead of CT scanning for staging iCC within the liver” (strong recommendation) [doi: 10.1016/j.jhep.2023.03.010] since MRI was superior to CT in staging iCC for T1B, T2, and even T3/T4 tumours [doi: 10.1002/hep4.1774.]. Please revise this sentence.

Re: According to reviewer suggestion, we have revised the text (see lines 135-143)

Lines 232-234. MRCP is pivotal also to understanding the anatomy of the biliary tree and its main variations and thus providing precise pre-operative information for surgeons in order to reduce the rate of procedure-related complications [doi: 10.1111/joa.13808].

Re: According to reviewer suggestion, we have additionally pointed out this aspect in the text  (see lines 359-367).

Lines 244-250. It is important to state that CEUS is an operator-dependent technique.

Re: According to reviewer suggestion, we have pointed out this aspect in the text  (see line 431).

Lines 251-252. The affirmation that “CEUS is recommended as the first-line imaging technique in the characterization of newly discovered focal liver lesion in non-cirrhotic livers” is not true. According to EASL guidelines [doi: 10.1016/j.jhep.2023.03.010], despite the first suspicion of iCC is usually raised on ultrasound, the benefit of contrast-enhanced ultrasound (CEUS) in iCC is controversial and this, as correctly stated, is especially true in the presence of underlying chronic liver disease [doi: 10.1111/liv.12124]. Similarly, according to AASLD guidelines, “Contrast‐enhanced US, although insufficient as the sole diagnostic modality, may be considered when CT or MRI is inconclusive” [doi: 10.1002/hep.32771]. Moreover, as reported by the latest EFSUMB guidelines Guidelines for Contrast Enhanced Ultrasound (CEUS) in the Liver [doi: 10.1055/a-1177-0530], CEUS is recommended for the characterization of focal liver lesions in the non-cirrhotic liver in patients with inconclusive findings at CT or MR imaging (strong recommendation) or if both CT and MR imaging are contraindicated (strong recommendation). The primary aim of CEUS in patients with a non-cirrhotic liver is to differentiate benign from malignant and, therefore, it is useful to facilitate the clinical decision as to whether a sonographically detected liver lesion needs further investigation (CT or MRI) or surgery. Please revise this concept.

Re: Following the suggestion of the reviewer we have better clarified this aspect in the text (see lines 431-444).

Please provide a Table reporting the imaging features of iCC with every discussed imaging technique.

Re: Following the suggestion of the reviewer we have summarized the main radiological featureas of iCC in Table 2.

Regarding Figure 1. Please provide an example of iCC in every imaging technique (CT, MRI, CEUS and PET), making sure to provide both arterial and venous phase images and, if possible, also those MRI sequences with ancillary findings that are highly suspicious for iCC (as hyperintensity in T2-w images and restricted diffusion).

Re: Following the suggestion of the reviewer we added examples of iCC in every imaging technique (see Figures 1-5).

Please also expand the role of PET scanning in iCC management, further underlying its usefulness in identifying lymph node metastasis.

Re: According to reviewer suggestion, we have pointed out this aspect in the text  (see lines 368-423).

From a stylistic point of view, I suggest discussing the current evidence regarding radiomics in a single and separate chapter rather than discussing it in at the end of each chapter. Since its use in clinical practice has not yet been implemented, I believe these changes will facilitate reader comprehension.

Re: Following the suggestion of the reviewer we added a paragraph regarding radiomics application (see lines 581-623).

Finally, I think references should be reformatted as suggested by the Author’s guidelines (Author 1, A.B.; Author 2, C.D. Title of the article. Abbreviated Journal Name YearVolume, page range)

Re: According to the reviewer observation we revised the references. We apologize for the blameworthy lack of attention during the written of the first draft.

Reviewer 2 Report

I have read with interest this manuscript that is a narrative simple review on radiological diagnostic features in intrahepatic cholangiocarcinoma. Overall, the paper is well written and readable. However, some changes should be provided:

- The abbreviations ICC and iCCA are both reported, while only one should be used. 

- There is no mention on lymph nodes, either loco-regional or distant, metastases. This point is very important when patients appear to be resectable. Please add this part for CT, and MRI.

- There is no mention on DWI in defining for instance lymph nodes status in comparison with FDG-PET

- The paragraph on PET is too short. Please improve that part. 

- While this is a simple narrative review, it would be nice to have a table with Se, Sp, PPV, NPV of the different techniques for the different study endpoint (i.e., parenchyma, vascular involvement, LN, distant mets, etc...).

Author Response

I have read with interest this manuscript that is a narrative simple review on radiological diagnostic features in intrahepatic cholangiocarcinoma. Overall, the paper is well written and readable. However, some changes should be provided:

- The abbreviations ICC and iCCA are both reported, while only one should be used.

Re: According to the reviewer observation we revised the text. We apologize for the blameworthy lack of attention during the written of the first draft.

- There is no mention on lymph nodes, either loco-regional or distant, metastases. This point is very important when patients appear to be resectable. Please add this part for CT, and MRI.

Re: Following the suggestion of the reviewer we additionally pointed out this aspect in the text for both CT and MRI (see lines 150-185, 283-297 and 304-318)

- There is no mention on DWI in defining for instance lymph nodes status in comparison with FDG-PET.

Re: Following the suggestion of the reviewer we additionally pointed out this aspect in the text (see lines 203-206, 214-216, 304-318 and 325-351)

- The paragraph on PET is too short. Please improve that part. 

Re: According to reviewer suggestion we expanded the section regarding PET (see lines 368-423)

- While this is a simple narrative review, it would be nice to have a table with Se, Sp, PPV, NPV of the different techniques for the different study endpoint (i.e., parenchyma, vascular involvement, LN, distant mets, etc...).

Re: Following the suggestion of the reviewer we have summarized the diagnostic accuracy of the different imaging techniques in Table 3.

Round 2

Reviewer 1 Report

The Authors addressed raised points appropriately.